# A New Nrf2 Inhibitor Enhances Chemotherapeutic Effects in Glioblastoma Cells Carrying p53 Mutations

**DOI:** 10.3390/cancers14246120

**Published:** 2022-12-12

**Authors:** Rayhaneh Afjei, Negar Sadeghipour, Sukumar Uday Kumar, Mallesh Pandrala, Vineet Kumar, Sanjay V. Malhotra, Tarik F. Massoud, Ramasamy Paulmurugan

**Affiliations:** 1Department of Radiology, Molecular Imaging Program at Stanford (MIPS), Canary Center at Stanford for Cancer Early Detection, Stanford University School of Medicine, 3155 Porter Drive, Palo Alto, CA 94305, USA; 2Department of Radiation Oncology, Stanford University School of Medicine, 3155 Porter Drive, Palo Alto, CA 94305, USA; 3Department of Cell, Development and Cancer Biology, Knight Cancer Institute, Oregon Health & Science University, Portland, OR 97201, USA; 4Center for Experimental Therapeutics, Knight Cancer Institute, Oregon Health & Science University, Portland, OR 97201, USA

**Keywords:** glioblastoma, Nrf2, p53, chemotherapy, small molecule compounds

## Abstract

**Simple Summary:**

Glioblastoma (GBM) is the most common and deadliest brain tumor. Currently, there is no successful treatment available for GBM patients. Different genes are mutated in GBM tumors, including the tumor suppressor p53 gene. Another important gene responsible for cellular homeostasis is Nrf2, which is upregulated in GBM. In this study, we investigate a set of Nrf2 inhibitors and activators identified from a library of small molecules for their roles in enhancing treatment outcome, and to understand the interplay between p53 and Nrf2 in GBM. We studied GBM cells genetically engineered to express clinically important mutants of p53. We were able to identify an Nrf2 inhibitor that significantly reduces cell growth and increases cellular apoptosis in GBM cells. We observed a synergistic therapeutic effect of combined Nrf2 inhibitor and chemotherapeutic drug temozolomide.

**Abstract:**

TP53 tumor suppressor gene is a commonly mutated gene in cancer. p53 mediated senescence is critical in preventing oncogenesis in normal cells. Since p53 is a transcription factor, mutations in its DNA binding domain result in the functional loss of p53-mediated cellular pathways. Similarly, nuclear factor erythroid 2–related factor 2 (Nrf2) is another transcription factor that maintains cellular homeostasis by regulating redox and detoxification mechanisms. In glioblastoma (GBM), Nrf2-mediated antioxidant activity is upregulated while p53-mediated senescence is lost, both rendering GBM cells resistant to treatment. To address this, we identified novel Nrf2 inhibitors from bioactive compounds using a molecular imaging biosensor-based screening approach. We further evaluated the identified compounds for their in vitro and in vivo chemotherapy enhancement capabilities in GBM cells carrying different p53 mutations. We thus identified an Nrf2 inhibitor that is effective in GBM cells carrying the p53 (R175H) mutation, a frequent clinically observed hotspot structural mutation responsible for chemotherapeutic resistance in GBM. Combining this drug with low-dose chemotherapies can potentially reduce their toxicity and increase their efficacy by transiently suppressing Nrf2-mediated detoxification function in GBM cells carrying this important p53 missense mutation.

## 1. Introduction

Glioblastoma (GBM) is the most malignant form of primary brain tumor, with an average survival time of 14.6 months after initial diagnosis [1]. Following surgery and radiation therapy, temozolomide (TMZ) is the current mainstay chemotherapy for GBM [2]. However, patients develop drug resistance over time, which results in cancer recurrence and progression [3,4,5]. In addition, aggressive drug regimens cause considerable toxicities and side effects without advancing therapeutic efficacy [6]. New therapeutic agents that sensitize GBM cells to current chemotherapy drugs, such as TMZ, at subtoxic doses could improve overall therapeutic outcomes [7,8,9].

Nuclear factor-erythroid factor 2-related factor 2 (Nrf2) is an important transcription factor that plays a major role in GBM therapy by negatively impacting chemotherapy through activation of endogenous phase II detoxification mechanisms. In normal cells, Nrf2 is expressed at physiological levels to maintain the cellular homeostasis and to prevent diseases including cancer [10]. Nrf2 protein is maintained in the cytoplasm as an inactive complex by binding to a repressor molecule known as Keap1 (Klech-like ECH-associated protein 1) [11]. During redox stress, the Nrf2-Keap1 pathway is activated. Nrf2 is phosphorylated by protein kinase C and translocases to the nucleus, where it binds to DNA binding domains of the phase-II enzyme to regulate their expression through antioxidant response elements (AREs) [12]. This process prevents reactive oxygen species (ROS)-mediated DNA damage, and maintains cellular homeostasis. However, previous studies have shown that somatic mutations to Nrf2 in cancer cells disrupt its interaction with Keap1 and prevent ubiquitination that leads to higher accumulation of Nrf2 in cells [13]. In cancer therapy, higher levels of Nrf2 with the upregulated antioxidant pathway lead to cancer cells acquiring protection from the cytotoxic effects of the chemotherapeutic drugs and the development of chemoresistance.

Tumor suppressor p53 is another important transcription factor necessary for the response of cancer cells to various cytotoxic therapies. p53 regulates cellular apoptosis by controlling the expression of genes involved in cell cycle arrest, DNA repair, apoptosis, and senescence [9]. *TP53* gene is mutated in 85% of GBMs and contributes to the loss of apoptotic response in cancer cells [14]. Many chemotherapeutic drugs, such as doxorubicin (DOX), activate p53-mediated biological mechanisms to induce cytotoxic cell death in cancer therapy [14]. In addition, mechanisms of inhibition of apoptosis by p53 owing to the upregulation of Nrf2 in cancer cells have been proposed. Thus, understanding the underlying molecular mechanisms of the crosstalk between these two proteins in response to chemotherapy in mutant-p53 GBM cells can shed light on the possibility of using Nrf2 inhibitors/activators to improve GBM treatment [15,16]. 

Here, we investigate the interplay between p53 and Nrf2 pathways after treatment using compounds that effectively alter the Nrf2 pathway. We use A2780 cells stably expressing the Firefly luciferase reporter gene under an antioxidant response element (ARE) for screening Nrf2 inhibitors. We use this cell line to study induction of Nrf2 protein upon treatment using small molecule drugs that have been recently studied by Eldhose et al. [17] (CET-CH-1 to CET-CH-5), along with a novel high potential anticancer agent that we synthesized (CET-CH-6) based on this compound library (Figure 1A). To study the relationship between mutant p53 and Nrf2, we engineer p53-null LN308 GBM cells to stably express clinically important p53 mutants (Figure 1B). Through in vitro experiments, we select an Nrf2 inhibitor that enhances the therapeutic effect of TMZ and DOX on GBM cell lines of different p53 status. Tumors of LN308 cells carrying a p53 mutation at amino acid 175, a common missense structural mutation with no effective treatment available, show significant inhibition in growth in vitro and in vivo to the selected Nrf2 inhibitor when combined with TMZ.

## 2. Materials and Methods

### 2.1. Chemicals, Enzymes, and Reagents

We purchased A2780 human ovarian cancer cells, U87-MG (#HTB-14), HEK293T (#CRL-3216), and MDA-MB-231 (#CRM-HTB-26) from ATCC (Manassas, VA, USA). LN308 cells (p53 null) were a kind gift from Markus Weiler (Heidelberg University, Germany). We purchased propidium iodide (#P4864), TBHQ (#112941), and doxorubicin (D1515) from Sigma-Aldrich (St. Louis, MO, USA), and DMSO (D159-4) from Fisher Scientific (Santa Clara, CA, USA). We used FBS (#26140087), Penicillin-Streptomycin-Glutamine (#10378016), and sodium bicarbonate (#25080094) from GIBCO BRL (Frederick, MD, USA). We used plasmid extraction kits and DNA gel elution kits from Qiagen (Valencia, CA, USA) and Epoch Life Sciences (Missouri City, TX, USA). We purchased D-Luciferin from Biosynth (Staad, Switzerland). Antibiotics for bacterial and cell culture experiments were from Sigma (St. Louis, MO, USA), and bacterial culture media was from Difco (Franklin Lakes, NJ, USA).

### 2.2. Synthesis of the Compounds

Synthesis and characterization of chalcone derivatives (CET-CH-1 to CET-CH-5) were performed as described previously [17]. Synthesis of CET-CH-6 is described in the Appendix A.

### 2.3. Cell Culture

We cultured the cells in high-glucose Dulbecco’s Modified Eagle’s Medium (#11965092) (Life Technologies, Carlsbad, CA, USA) supplemented with 10% FBS and 100 U/mL penicillin and 100 μg/mL streptomycin solution. We maintained cells at 37 °C in 5% CO_2_, and plated them for experiments to an optimal confluency during their exponential growth phase.

### 2.4. Construction of Plasmid and Lentiviral Vectors Expressing Nrf2-ARE-FLuc Sensor

We constructed an Nrf2-ARE-FLuc biosensor using a standard PCR based cloning strategy. In brief, we first constructed ARE from Nrf2 target gene (*NQO1*) by replacing the CMV promoter from pcDNA 3.1+ vector backbone. Then, we introduced the full-length luciferase reporter gene downstream of ARE in NheI and XhoI sites to achieve the pcDNA-NQO1-FLuc construct. The NQO1-FLuc fragment released from the pcDNA-NQO1-FLuc construct using SpeI/XhoI was subcloned into pHAGE lentiviral backbone for producing lentiviruses to create A2780 stable cells for screening Nrf2 activators/inhibitors. We used the sequenced vector constructs (confirmed by Sequetech, Mountain View, CA, USA) in cell culture transient transfection experiments to produce lentivirus. We used the three-vector based transfection system described later for lentivirus production and further transduction in creating stable cells.

### 2.5. Construction of Plasmid and Lentiviral Vectors Expressing p53 Variants

We constructed plasmid vector expressing p53 proteins of wildtype (wt) and mutants under a ubiquitin promoter in a pcDNA 3.1 (+) backbone with a puromycin antibiotic selection marker. We first constructed a vector with p53-wt protein using a standard PCR based cloning strategy. We used site-directed mutagenesis to create a single amino acid change at positions 175 (p53^R175H^), 220 (p53^Y220C^), 245 (p53^G245S^), and 282 (p53^R282W^) using a site-directed mutagenesis kit from Stratagene (Agilent Technologies, Santa Clara, CA, USA). We used the sequence confirmed vector constructs for transient transfection experiments and for sub-cloning into a lentiviral backbone. These five constructs were further released along with the constitutive Ubiquitin promoter (Ubiquitin-p53) and sub-cloned into a pHAGE-lentiviral backbone at the SpeI/XhoI restriction enzyme sites. We then chose the sequence confirmed vectors for lentiviral production using the three-vector transfection system as described later.

### 2.6. Lentiviral Production and Stable Cell Generation by Lentiviral Transduction

We maintained A2780 and LN308 cells in DMEM-high glucose medium with 10% FBS, 100 U/mL penicillin, and 100 μg/mL streptomycin. For lentiviral production, we used HEK293-FT cells grown in Dulbecco’s Modified Eagle’s Medium supplemented with 10% FBS, 100 U/mL penicillin, and 100 μg/mL streptomycin. We used a three-vector transfection system (pHAGE + VPR + VSVG) to produce lentivirus for NQO1-ARE-FLuc and p53 variants with a GFP biosensor expressing p53^wt^, p53^175^, p53^220^, p53^245^, and p53^282^ in a 3:2:1 ratio using a standard calcium phosphate transfection method. Twenty-four hours later, we washed the cells once with PBS and replaced with 8 mL of complete medium containing 10 mM Hepes buffer, pH 7.5. We enriched the virus from the medium after 48 h by filtration followed by ultracentrifugation. After titration, we used viruses for creating the A2780-Nrf2-ARE-FLuc stable clone and LN308 cells stably expressing p53 variants. In brief, we transduced the cells plated to 80% confluence in 10-cm plates 24 h before transduction. We washed the cells once with PBS, and then added 2 mL of 1 × 10^8^ PFU virus mixed with 3 mL of serum-free Opti-MEM and 5 μL of 1 mg/mL polybrene. Each plate was incubated for 4 h at 37 °C and 5% CO_2_ with intermittent mixing. Four hours later, we washed the plates once with PBS and supplemented with 10 mL of complete medium containing 10% FBS, 100 U/mL penicillin, and 100 μg/mL streptomycin. We sub-cultured the cells twice before we selected stable cells using FACS sorting. The clonal populations of cells expressing equal levels of p53 constructs were used in further experiments. 

### 2.7. Evaluation of Nrf2 Activation Using Are Signaling Characterized by FLuc Biosensor Bioluminescence Imaging

We treated A2780 cells stably expressing NQO1-FLuc biosensor using different Nrf2 modulators (CET-CH-1 to CET-CH6) along with TBHQ as a positive control, and DMSO as a solvent negative control. We used CET-CH-1 to CET-CH-6 in various concentrations (0, 1.25, 2.5, 5,10, 20, and 30 mM) for the screening studies. We plated 5000 cells/well in 96-well black walled plates 24 h before treating the cells using these Nrf2 modulators. After 48 h, we washed the plates once with PBS and added 30 μg of D-Luc (30 mg/mL) in 50 μL of PBS to each well using a multichannel pipette. We imaged the plates by continuous acquisition of 1 min integration times for a total of 15 min. We quantified luciferase signal from each well using Living Image software by drawing regions of interest (ROIs) over each well. We plotted the relative signal in response to treatment as a graph. The FLuc signal was proportional to Nrf2 activation. 

### 2.8. Immunoblot Analysis for the Evaluation of Downstream Target Gene Signaling

For the analyses of each corresponding pathway, we collected cells after appropriate treatment by centrifuging at 5000 rpm for 5 min and lysed the cell pellets in RIPA buffer containing protease inhibitor cocktail and EDTA by sonication three times at 40% amplitude for 15 s each in ice. We centrifuged the cell lysates at 15,000 rpm for 15 min at 4 °C to remove any insoluble proteins. We estimated the protein concentration using Nanodrop 2000 (Thermo Scientific, Waltham, MA, USA) and further confirmed this using a BCA protein assay system from BioRad. We prepared 200 μg of total protein in 1× Lamelli loading buffer with 5% β-mercaptoethanol (Life Technologies, Carlsbad, CA, USA), and denatured this at 95 °C for 5 min. We resolved the samples on 4-12% SDS-polyacrylamide pre-cast gels (Life Technologies, Carlsbad, CA, USA) and transferred to a polyvinylidene difluoride nylon membrane (Bio-Rad, Hercules, CA, USA) using wet electroblotting. Membranes were blocked in 5% non-fat dry milk in Tris-buffered saline containing 0.05% Tween-20 (TBST), (#P9416) from Sigma-Aldrich followed by incubation with a primary antibody. As appropriate, we used antibodies suitable for the detection of genes suggested by the manufacturers. Following three TBST washes, we incubated membranes with the appropriate horseradish peroxidase-conjugated secondary antibodies (Sigma Aldrich, St Louis, MO, USA). After three additional TBST washes, we incubated immunoblots with the Pierce ECL Western Blotting Substrate (Thermo Scientific, Waltham, MA, USA) for 1 min, and then collected the chemiluminescence signal using an IVIS Lumina imaging system (PerkinElmer, Bridgeville, PA, USA). We used a GAPDH (#5174) and Lamin B1 (#12586) primary antibodies (CST, Danvers, MA, USA) diluted in PBST at 1:2000 as a loading control house-keeping antibody for cytoplasmic and nuclear, respectively. For the expression of genes in response to CET-CH-6 (5 μM) and DOX (0.5 μM) in different treatment conditions, we used Nrf2 (#sc-365949) and Keap1 (#sc-365626) from Santa Cruz Biotechnology (Santa Cruz, CA, USA) as primary antibodies by diluting in PBST at 1:1000. For the expression of p53 in LN308 cells stably expressing p53 variants, we used p53 primary antibody from Cell Signaling (#9282, CST, 1:2000). Similarly, for measuring the change in response to treatment from LN308 cells stably expressing p53 variants after treating with CET-CH-6 and DOX, we used p53 (#9282), Bcl2 (#4223), Bax (#2774), Noxa (#14766), Puma (#4976S), and p21 (#2947) from Cell Signaling (CST, 1:2000), and SOD2 primary antibody from Santa Cruz Biotechnology (#sc-137254). In MDA-MB-231 cells, where we investigated the distribution of the proteins in the nucleus and cytoplasm, we used a different extraction kit by following the manufacturer’s protocol. We quantified nucleocytoplasmic distribution of Nrf2 and Keap1 proteins upon different treatment conditions using a Thermo Scientific NE-PER^TM^ Nuclear and Cytoplasmic Extraction Reagents kit (# 78833, Thermo Scientific^TM^). Briefly, cells from different treatment groups were collected and pelleted by centrifugation. The cells were suspended in ice-cold CER II. We vortexed the tubes and centrifuged the samples and transferred the supernatant that contained a cytoplasmic extract. The pellet which contained the nuclei was suspended in NER. We vortexed the tubes (4 times with 10 min interval) and centrifuged at maximum speed and immediately transferred the supernatant (nuclear extract). The protein fractions were used for immunoblotting analysis.

### 2.9. Confocal Microscopy

We used confocal microscopy (TCS SP8, Leica, Wetzlar, Germany) to image MDA-MB-231 cells to study the mechanism of activation of Nrf2 in response to CET-CH-6. We seeded 100,000 cells on cover slips (Fisher Scientific, Santa Clara, CA, USA) placed in 6-well plates 24 h before treatment. We treated the cells with CET-CH-6 at 2.5 μM concentration for 24 h. On the day of imaging, we washed the cells with PBS, and fixed them in 1% ice cold PFA by keeping them for 20 min on ice. We then washed them twice with ice cold PBS. We permeabilized the cells by adding 1 mL of permeabilization solution for 1 h. We stained the cells using Nrf2 and Keap1 antibodies (1:200) diluted in PBST with 1% bovine serum albumin and incubated them overnight in a humidified chamber. The next day, we aspirated the solution and washed it with PBST three times, and then stained it with anti-mouse FITC secondary antibody (1:200) for 2 h at room temperature. We then washed the cells with PBST three times, and added Hoechst 33342 to stain the nucleus, mounted cells, and acquired optical images using appropriate imaging channels. We also confirmed the expression of GFP in LN308 clones using confocal microscopy by culturing them on coverslips and imaging them in a GFP channel.

### 2.10. Evaluation of Apoptosis in U87-MG-p53^wt^ and LN308 Cells Stably Expressing p53 Variants in Response to Treatment with Nrf2 Modulators in the Presence of Chemotherapy Using Propidium Iodide Staining Based Flow Cytometry Analysis

For apoptosis induction in U87-MG and LN308 cells stably expressing different p53 variants, we seeded the cells at 80% confluency in 12-well plates at 1 × 10^5^ cells/well in DMEM medium supplemented with 10% FBS for 24 h before co-treatment with CET-CH-1 to CET-CH-6 and DOX. We then treated cells with different concentrations of CET-CH-1 to CET-CH-6 (0, 1.25, 2.5, and 5 μM) in the presence and absence of DOX and incubated them further for 48 h. For the assessment of apoptotic populations, we used two different assays. In the first assay, we collected and fixed the cells for PI-based flow cytometry analysis. In this assay, the apoptotic population is distinguished from the non-apoptotic cells based on the DNA content and its locational and cellular distribution. In non-apoptotic cells, DNA is localized within the nucleus, whereas, in apoptotic cells, the DNA is fragmented. Since we are fixing the cells, the cell membrane is permeable for PI independent of their cellular status. Hence, after PI staining, the apoptotic population shows a weak PI signal from the fragmented DNA while non-apoptotic cells show a strong signal from the intact DNA [18]. We collected the cells by trypsinization, resuspended them in 0.5 mL ice-cold PBS, and then fixed them by adding 2 mL of 100% ice-cold ethanol while vortexing the sample to achieve a final concentration of ethanol to 70%. We stored the samples at −20 °C until FACS analysis. We have successfully used this assay in many of our previous publications [19,20,21,22]. Briefly, the cells were collected by centrifugation, washed once in PBS, and re-suspended in 0.5 mL PBS/RNase A (10 µg/mL)/Triton X-100 (0.1%) buffer containing 0.5 µg/mL of PI. After 15 min of incubation at room temperature in the dark, we washed the cells once with PBS and FACS analyzed using a Guava cytometer. In the second assay, we treated the cells in a similar way, but we stained the cells using the Annexin-V/PI kit (Invitrogen, Waltham, MA, USA) right after the collection of cells. We assayed the samples using the Guava cytometer. We analyzed the generated results using FlowJo 8.6.6 software for live and dead cell analysis.

### 2.11. Evaluation of Cell Viability Using MTT Assay

We used MTT assays to assess the viability of cells after exposure to different treatment conditions. We seeded the cells in 96-well plates at a density of 2500 cells/well 24 h before treatment. We then treated cells with different concentrations of CET-CH-6 (0, 0.125, 0.25, 0.5 μM) in the absence and presence of DOX (0.5 μM), and TMZ (250 μM). At 24, 48, and 72 h after treatment, we aspirated the DMEM, and added 50 μL of MTT solution (diluted in phenol red free RPMI to a final concentration of 1.25 μg/mL) per well of 96-well plates. We then centrifuged plates at 1000 rpm for 3 min before incubating with the MTT solution at 37 °C in the dark for 2 h. After 2 h, we carefully removed the MTT solution without disturbing the formazan crystals and added 100 μL DMSO per well, and allowed for dissolving by keeping it in the incubator for 30 min. After incubation, the spectroscopic signal at 565 nm wavelength was measured on a Tecan machine in accordance with manufacturer instructions. The results were given as percentages of viable cells relative to the control condition.

We also calculated drug synergy by the coefficient of drug interaction (CDI): CDI = AB/(A × B). Based on the absorbance of each group, AB is the ratio of the combination group to control group, and A or B is the ratio of the single agent group to control group. CDI of less than 1 represents synergistic effects.

### 2.12. In Vivo Evaluation of Co-Treatment of CET-CH-6 and TMZ in Subcutaneous GBM Models in Balb/C Nude Mice

All animal experiments were performed in accordance with the Stanford University Administrative Panels on Laboratory Animal Care. We used NSG mice aged 4–6 weeks. We conducted two sets of animal studies. In the first set, we implanted mice with 5 × 10^6^ LN308-p53^wt^ and LN308-p53^R175H^ into the left and right flanks, respectively. We allowed tumors to grow until they reached 3 mm in diameter, and randomized the mice into four groups (number of tumors, *n* = 4–6 per group). Control mice did not receive any treatment, the TMZ treated group received TMZ (12.5 mg/kg body weight), the CET-CH-6 group received CET-CH-6 (2.5 mg/kg body weight), and the CET-CH-6 + TMZ group received a co-treatment of those drugs at similar concentrations. We treated mice with intraperitoneal injection of drugs diluted in 10% PEG-400 once every two days for five cycles of treatment. We recorded mouse body weight and tumor volume throughout the study. We sacrificed the animals when tumor size exceeded the allowed limit (1.5 cm^3^), or at day 75 after start of the treatment. We then collected organs and tumors for ex vivo histological, toxicity, and apoptosis analysis. Based on the outcome of the first set, we implanted another set of animals with U87-MG-p53^wt^ and LN308-p53^R175H^ cells in the right and left flanks, respectively, and repeated the same treatment plan. We used TUNEL staining to measure the induced apoptosis.

### 2.13. H&E Staining of the Organs

We harvested the mice organs and fixed them in 4% paraformaldehyde overnight at 4 °C, and immersed them in 70% ethanol. We sent the samples to the Stanford Animal Histology Services for H&E staining, and imaged them using a Nanozoomer (Hamamatsu, Japan).

### 2.14. TUNEL Staining of Tumor Tissues

To quantify tumor cell apoptosis upon treatment, we performed a TUNEL assay (TACS 2 TdT-DAB In Situ Apoptosis Detection Kit, Trevigen, Gaithersburg, MD, USA). We sliced paraffin embedded sections and deparaffinized them to perform the assay, labeled according to the manufacturer’s instructions. We scanned the slices under microscope and performed image analysis using ImageJ and the Color Threshold and Particle Analysis functions. We presented the total number of TUNEL positive cells in each sample per field of view to quantitively measure the treatment outcome.

### 2.15. Statistical Analysis

We performed statistical analyses and prepared graphical presentations using Prism 9 (GraphPad). Results were presented as mean ± SD. Results were representative of at least three independent experiments. Observations with a *p*-value of less than 0.05 were considered statistically significant. Kaplan–Meier analyses were used for statistical analyses of survival rates.

Note: Additional methods, please refer to Appendix A.

## 3. Results

We show the complete experimental workflow as a schematic in Figure 1.

### 3.1. Synthesis

Chalcone analogues CET-CH-1 to CET-CH-5 (Appendix A) were synthesized and characterized as described previously [17,23,24,25]. CET-CH-6 is a novel chalcone analogue, which was prepared according to the previously reported procedure for a similar analogue [26], by the Claisen–Schmidt condensation reaction between 1-(5-hydroxy-2-methoxyphenyl)ethan-1-one and 3-formylbenzonitrile as described in the Appendix A.

### 3.2. CET-CH-2 and CET-CH-6 Were Identified as Nrf2 Activator and Inhibitor, Respectively

To identify new compounds that can efficiently change the transcriptional activity of Nrf2 protein, we engineered A2780 ovarian cancer cells to express the Firefly luciferase (FLuc) reporter gene under an ARE response element derived from NQO1, the downstream target gene of Nrf2. We used lentivirus transduction to stably engineer A2780 cells using this construct. We FACS sorted the cells to enrich the clonal population of cells stably expressing NQO1-FLuc sensor for further expansion to measure Nrf2 activity in response to its modulators. We plated cells in various formats for screening and further confirmation of Nrf2 activation by indirectly measuring NQO1-FLuc expression. In brief, we treated the cells with CET-CH-1 to CET-CH-6 for 24 h and assessed for the induced FLuc expression using an IVIS optical imaging system (Figure 2A). We used DMSO, and TBHQ (a known chemical activator of Nrf2) as negative and positive controls, respectively, to measure the specificity of Nrf2 activation. We initially treated the cells with six different compounds in the dose range of 10 to 30 μM (Figure 2B). We observed enhancement in the expression of FLuc in cells treated with CET-CH-1, CET-CH-2, CET-CH-3, and CET-CH-4, while finding significant reductions in the expression in cells treated with CET-CH-5 at a concentration of 20 (*p*-value < 0.01) and 30 μM (*p*-value < 0.001), and CET-CH-6 at concentrations ranging from 10 to 30 μM (*p*-value < 0.001). Among the activators, CET-CH-2 resulted in the highest level of activation, which was about 4-fold at 10 μM (*p*-value < 0.0001). Similarly, among the inhibitors, CET-CH-6 showed the highest level of reduction in FLuc signal, which was about 3-fold at 30 μM (*p*-value < 0.0001) compared to DMSO control. In addition, we observed significant levels of cell death at these higher doses owing to toxicity. Thus, we reduced the dose of these compounds and tested them in the range of 1.25 to 10 μM concentrations (Figure 2C). We found a dose dependent activation of FLuc signal from CET-CH-2, while CET-CH-6 yielded a dose dependent reduction in signal (Figure 2D). Based on these observations, we selected CET-CH-6 at 5 μM for further downstream pathway analyses.

### 3.3. Mechanism of Nrf2-Keap1 Signaling Pathway in Response to CET-CH-6

To understand the mechanism of inhibition of Nrf2 by CET-CH-6, we performed immunofluorescence imaging using confocal microscopy and immunoblot analysis for the overexpression of Nrf2 using MDA-MB-231 cells (Figure 3A). We selected MDA-MB-231 cells because this cell line shows very high levels of Nrf2 expression [27]. We quantified the level of cytoplasmic and nuclear Nrf2 and Keap1 by normalizing the data to the total amount of expressed protein. Quantification of the distribution of Nrf2 and Keap1 before and 24 h after treatment using CET-CH-6 showed that both cytoplasmic Nrf2 and Keap1 increased (*p*-value < 0.05), after treatment (Figure 3B). We observed overall reduction in the total amount of Nrf2 and Keap1 expressed in cells after treatment with CET-CH-6. The immunoblot results of treatment with CET-CH-6, TMZ, and the combination of them are shown in Figure 3C. We quantified the expression of cytoplasmic and nuclear Keap1 and Nrf2 by normalizing them to GAPDH and Lamin B1, respectively. In Figure 3D, the Nrf2 signals are shown that are normalized to the control cells. After treatment of the cells with CET-CH-6, the amount of Nrf2 in cytoplasm and nucleus increased but not significantly. We observed a decrease in cytoplasmic Nrf2 and an increase in nuclear Nrf2 after co-treatment with CET-CH-6 and TMZ. Keap1 changes were more significant after treatment with CET-CH-6 and the combination of CET-CH-6 and TMZ (Figure 3E). In both of these conditions, the expression of cytoplasmic Keap1 decreased (*p*-value < 0.05). We repeated this experiment by including CET-CH-2, an Nrf2 activator, for a comparative evaluation. We observed that treatment of cells with CET-CH-2 did not alter the expression of Keap1 neither in the nucleus nor cytoplasm. However, CET-CH-2 treatment increased the expression of Nrf2 in cytoplasm but decreased it in the nucleus. Co-treatment of cells with CET-CH-2 and TMZ reduced (*p*-value < 0.05) the expression of Keap1 and Nrf2 in both cytoplasm and nucleus (Appendix A). 

### 3.4. Engineering of p53-Null LN308 Cells to Stably Express p53 with Clinically Important Structural Mutants to Study Nrf2 Inhibitors in Combination Chemotherapy

The p53 tumor suppressor gene is mutated in more than 85% of GBMs, and its interaction with Nrf2 regulates cellular apoptosis and drug resistance. Hence, we explored the relationship between p53 and Nrf2 alterations in response to CET-CH-6 and TMZ. Since there is no GBM cell line available with different p53 status in a single genetic background, and GBM cells are highly heterogenous, we generated GBM cell lines with different p53 structural mutations in the DNA binding domain in p53-null LN308 cells. We used a lentivirus-based transduction system to engineer these cells. We used a lentivirus expressing GFP along with p53 under dual Ubiquitin promoters to engineer, enrich, and clonally expand cells using FACS sorting. LN308 cells were independently transduced with five different lentiviruses, which express p53^wt^, p53^R175H^, p53^Y220C^, p5^G245S^, and p53^R282W^ (Figure 4A). We used the co-expressed GFP to sort and enrich clonal population of cells (Figure 4B). We also confirmed the expression of GFP in flow cytometry, where more than 90% of cells in each clone had high expression of GFP (Figure 4C). We used clonal populations of cells with equal GFP levels to quantify p53 levels using immunoblot analysis. Even though we had equal amounts of GFP expression, owing to variations in protein stability of p53 structural mutants, we observed different amounts of p53 proteins in cells expressing different variants (Figure 4D). We also tested cells with different mutants of p53 for their proliferation using an MTT assay at 24, 48, and 72 h after seeding (Appendix A). We found that cells with p53^wt^, p53^220^, and p53^245^ have similar proliferation patterns to the parental LN308 cells (i.e., around 40% increase in the cell number after 48–72 h). In contrast, LN308-p53^175^ had the slowest growth at about 10% increase in the cell numbers after 48–72 h. LN308-p53^282^ had about 75% increase in cell populations after 48 h, but the cell viability decreased to the initial percentage at 72 h after plating owing to confluency-associated growth inhibition. We treated these clones to observe any change in the expression levels of endogenous Keap1 using immunoblot analysis of cells in the presence and absence of CET-CH-6 (2.5 μM). For normalized Keap1, there was no significant change in the expression levels of Keap1 at baseline levels of different clones (Figure 4D). We used these clones for further in vitro and in vivo studies. When these clones were treated with CET-CH-6, Keap1 level decreased in LN308-p53^wt^ and LN308-p53^245^ clones (*p*-value < 0.05). However, treatment did not significantly change Keap1 levels in parental LN308, LN308-p53^175^, LN308-p53^220^, and LN308-p53^282^ cells (Figure 4E,F). 

### 3.5. CET-CH-6 Synergistically Enhances the DOX-Mediated Apoptotic Effects in GBM Cells

The inhibition of Nrf2-mediated detoxification mechanisms during chemotherapy in cancer cells can enhance treatment outcomes in cancer therapy. DOX is not used for the treatment of GBM owing to the blood–brain barrier reducing its bioavailability in the brain, and dose limited toxicity to normal tissues. Hence, we hypothesized that the use of CET-CH-6 can sensitize cancer cells to subtoxic doses of DOX and could therefore change GBM therapy options in the clinical setting. To evaluate this property of CET-CH-6, we co-treated GBM cells with CET-CH-6 and chemotherapeutic drugs to ultimately measure the sensitivity of cancer cells to low-dose chemotherapy. DOX is a highly toxic DNA damaging agent used to kill rapidly proliferating cells in cancer therapy. To test the selective enhancement of treatment effects using CET-CH-6, we treated U87-MG GBM cells expressing p53^wt^ with two of Nrf2 activators (CET-CH-1 and CET-CH-2) and inhibitors with (CET-CH-5 and CET-CH-6), along with DOX. Apoptotic cell populations are shown in Figure 5A, and the gating on FACS results is shown in Appendix A. We found no significant effects on the apoptotic induction by using CET-CH-1 or CET-CH-2 at different doses (1.25, 2.5, and 5 μM). In contrast, we observed a dose-dependent effect with ~8% increase in the apoptotic population after treating U87-MG cells with CET-CH-6 at a 5 μM dose (*p*-value < 0.01). CET-CH-5 also showed significant increase in cell apoptotic population at a 5 μM dose (*p*-value < 0.05).

We then treated different p53 variants of LN308 cells with CET-CH-6 (0, 1.25, 2.5, and 5 μM) and DOX (0.5 μM). Apoptotic cell populations are shown in Figure 5B, and the gating on FACS results are shown in Appendix A. In parental LN308 cells (p53-null), we observed that treatment with CET-CH-6 alone showed a dose-dependent increase in the apoptotic population, with the highest being 52% at 5 μM. In addition, co-treatment of LN308 cells using CET-CH-6 (5 μM) and DOX (0.5 μM) increased the apoptotic population from 48% to 57%, compared to the DOX treatment alone (*p*-value < 0.0001). On the other hand, in LN308 cells expressing p53 wild-type or other mutant variants, we observed increases in apoptotic populations by about 36% after treatment using CET-CH-6 alone. When we co-treated LN308-p53^wt^ with CET-CH-6 and DOX, we did not observe significant increase in apoptosis. Among LN308 cells with mutant p53, LN308-p53^175^ showed the highest apoptotic population (50%) when co-treated with CET-CH-6 and DOX, which was about 22% more than DOX alone treated cells. Other LN308 mutant cells (wt, 245, 282) had about 45% apoptotic populations using CET-CH-6 and DOX (Appendix A).

We repeated this assay using Annexin-V/PI staining to show the live and apoptotic populations. In this assay, the apoptotic cells are PI positive, while the live cells due to their intact membrane show a PI negative signal. Since our cells were engineered using lentivirus coding for p53 and GFP, and most of the Annexin-V/PI staining kits use Annexin-V tagged with fluorophores in the green channel (to avoid overlapping with the fluorescence from PI), we only showed the double staining results on an LN308-null parental cell line (without GFP expression) in Appendix A, while analyzing other cells only in PI fluorescence window after being stained using the same Annexin-V/PI staining kit and showed the results in Appendix A. For the rest of the clones, we only compared PI staining results on cells stained immediately after collection and the one using the fixed cells. There was a significant correlation between the two assaying methods (R^2^ = 0.87), and we observed similar levels of apoptotic cells using the Annexin-V/PI staining assay. 

Similar to the FACS results, we tested cell growth inhibition in an MTT assay. We treated cells with sub-toxic doses of DOX (0.5 μM) along with different concentrations of CET-CH-6 (0, 1.25, 2.5, and 5 μM). The results showed significant synergistic effects on the inhibition of cell proliferation to DOX when combined with different concentrations of CET-CH-6. Particularly, in co-treatment of DOX with CET-CH-6, the effects were more prominent for CET-CH-6 at concentrations of 2.5 and 5 μM. LN308-p53^175^ and LN308-p53^wt^ responded well to the co-treatment of CET-CH-6 at 2.5 and 5 μM concentrations (Figure 6A). CET-CH-6 also induced cell growth inhibition without supplementing DOX, but only at high concentrations. We got ICD of 0.51, 0.56, and 0.73 for 1.25, 2.5, and 5 μM concentration of CET-CH-6 at 72 h for LN308-p53^175^. Full graph of response to treatment at 24, 48, and 72 h assessed by MTT after DOX treatment is shown in Appendix A.

### 3.6. Treatment of Cells Using CET-CH-6 Synergistically Enhanced the Therapeutic Effect of TMZ Chemotherapy in GBM Cells

Since TMZ is a clinically used drug for GBM therapy, we evaluated CET-CH-6 in combination with TMZ in GBM cells of different p53 status. We used LN308 clones with different p53 variants for the purpose. We treated cells with sub-toxic doses of TMZ (250 μM) along with different concentrations of CET-CH-6 (0, 1.25, 2.5, and 5 μM), and performed an MTT assay to assess effects. The results showed significant synergistic effects on the inhibition of cell proliferation by TMZ when combined with different concentrations of CET-CH-6. We observed ICD of 0.43, 0.28, and 0.18 for 1.25, 2.5, and 5 μM concentration of CET-CH-6 at 72 h for LN308-p53^175^, which shows the synergistic effect of CET-CH-6 on TMZ. Unlike the DOX treatment study, when we treated different LN308 p53 variant cell lines with TMZ alone, we did not observe any significant change in the proliferation of those cells. However, co-treatment of LN308 cells with TMZ and CET-CH-6 (Figure 6B) showed significant inhibition in cell growth in LN308 cell lines expressing p53^wt^ (*p*-value < 0.0001), p53^175^ (*p*-value < 0.001), and p53^245^ (*p*-value < 0.0001) at 72 h post treatment. Full graph of response to treatment at 24, 48, and 72 h assessed by MTT after TMZ treatment is shown in Appendix A. This cell growth inhibition seemed more attributable to CET-CH-6 than TMZ, since the same cells showed similar levels of inhibition in cell proliferation when treated using CET-CH-6 alone. Hence, we used this combination for in vivo evaluation in mouse models of GBM. 

### 3.7. Immunoblot Analysis Supports the Differential Apoptotic Response of LN308 Clones Expressing Different p53 Variants Compared to Parental Cells to CET-CH-6 in Combination with DOX by Activating p53 and Nrf2 Pathway Proteins

To determine the influence of Nrf2 inhibition when using CET-CH-6 on proteins involved in apoptosis, DNA damage repair, and p53 and Nrf2 pathways, we tested LN308 cells engineered to express different p53 variants (p53^wt^, p53^175^, p53^245^, p53^282^), along with parental LN308 cells, for their differential response to treatment impact on important pathway proteins (Figure 7). We compared the expression level of each protein normalized to their housekeeping protein, GAPDH. We found that the p53 protein was stabilized in all engineered mutants compared to parental LN308 cells as we had shown in Figure 4D. p53 protein levels increased after treatment using DOX, and a combination of DOX and CET-CH-6. Both CET-CH-6 and DOX contributed to the increased p53 levels, but DOX treatment alone increased the p53 level slightly more than CET-CH-6 treatment alone. In LN308-p53^175^ cells, CET-CH-6 alone increased the p53 level to the same level as the DOX treatment alone. Treatment of GBM cells with CET-CH-6 in combination with DOX altered several p53 pathway proteins (Bcl2, BAX, p21, Puma, and Noxa). Bcl2 regulates cell death by inducing or inhibiting apoptosis, and BAX is a member of this family, being the negative regulator of Bcl2. These two proteins have opposite regulatory functions in cells. Bcl2 expression increased after CET-CH-6 and DOX combination treatment in cells with p53^wt^, p53^175^, and p53^282^, but not in p53^245^. P21 is a direct target of p53, and it is activated in cells as a cytostatic response. We observed higher expression of p21 in cells expressing p53^wt^, p53^175^, and p53^245^ after DOX and CET-CH-6 co-treatment. Puma is a p53 target protein, which regulates apoptosis and cell cycle arrest. Noxa is also a proapoptotic member of the Bcl2 family regulated by p53. Noxa expression is highly upregulated in p53 mutant clones, and to a lesser degree in the wild-type cells. Noxa level did not change in the parental cells. Nrf2 had no expression in the parental cell line as well as untreated control cells possessing p53 mutations (except for LN308-p53^175^). However, its baseline expression itself was upregulated in cells with p53^wt^ and mutant clones, and further enhanced after treatment with CET-CH-6 and DOX. SOD2 is an antioxidant marker of the Nrf2 pathway. SOD2 is not present in control cells but is activated in response to CET-CH-6 and DOX treatments. Treatment of parental and mutated LN308 cells with CET-CH-6 in combination with DOX increased SOD2 expression, but we did not see this effect in p53^wt^ cells. This suggests that this treatment was able to increase apoptotic and cytostatic responses based on the proteins involved in these pathways (p21 and Bcl2 upregulation), but it was unable to completely knockdown the expression of Nrf2 pathway because SOD2 remains expressed.

### 3.8. In Vivo Evaluation of CET-CH-6 and TMZ Combination Treatment in Tumor Xenografts of Glioblastoma Cells Expressing p53^wt^ (U87-MG and LN308-p53^wt^) and Mutant Phenotypes

Based on our in vitro results, we found that LN308-p53^175^ cells were very responsive to CET-CH-6 treatment alone, as well as in combination with DOX or TMZ in apoptosis and cell viability and proliferation assays, and in immunoblot analysis for various pathway proteins. To further evaluate the in vivo therapeutic effectiveness of CET-CH-6 as an Nrf2 inhibitor on cells expressing p53^mt^ phenotypes, we co-treated the animals bearing U87-MG, LN308-p53^wt^, and LN308-p53^175^ phenotypes for ten cycles of CET-CH-6 and TMZ co-treatment by following the schedule (Figure 8A). Our survival analyses of mice with LN308-p53^175^ showed that mice that received TMZ and the combination of TMZ plus CET-CH-6 had a significantly longer survival time than the control and CET-CH-6 alone groups (*p*-value < 0.05). Even though we did not observe significant improvement in the survival rate in combination treatment group compared to the TMZ alone group (Figure 8B), tumor growth after the last dose was slower in the combination group. We sacrificed all the mice on day 75. The tumor volumes are plotted for mice until the date that the first mouse was sacrificed, day 27 (Figure 8C–E). Overall, we observed a significant reduction in tumor volume in mice treated using TMZ, and the combination of TMZ and CET-CH-6, compared to the control group for tumors of all different cell lines. For LN308-p53^175^ and U87-MG cell lines, the reduction in tumor sizes due to treatment with TMZ and the combination happened on day 12. By day 27, about the last day of treatment, tumors have shrunk to their initial size (pre-treatment) or even smaller. For U87-MG cells, we see that both TMZ and combination therapy have eliminated the tumors. Tumor growth in mice treated with CET-CH-6 alone was not significantly different from the control group, including their survival results. This could be due to an insufficient dose of the drug or fast clearance of this small molecule. TUNEL staining showed a significant increase in the apoptotic populations when they received the combination treatment in LN308^175^ (*p*-value < 0.05) and LN308^p53^ (*p*-value < 0.05). In U87-MG, TUNEL staining showed an increase in apoptosis in the groups receiving TMZ or the combination of TMZ and CET-CH-6 (*p*-value < 0.05). The H&E staining showed no toxicity to any of the organs in the different treatment groups (Appendix A). 

## 4. Discussion

We identified a novel Nrf2 inhibitor, named CET-CH-6, that synergistically enhances therapeutic effects of TMZ and DOX in GBM cell lines with specific p53 hotspot mutations. To identify this inhibitor, we first developed a biosensor expressing the (NQO1-FLuc) reporter cell line, which indirectly measures the activation of the transcriptional activity of the Nrf2 protein. The testing of this sensor using a known Nrf2 activator, TBHQ, showed a specific activation (using FLuc expression) by 3- to 4-fold. We thus identified four compounds with significant Nrf2 activation while two others showed high and selective inhibition of Nrf2 activity by 4- to 5-fold (Figure 2). Further validation of these molecules in GBM cells with wild-type and mutant p53 showed differential expression of downstream target proteins of Nrf2. 

The presence of structural mutations in the DNA binding domain of p53 gene in GBM makes p53-augmented chemotherapy an attractive strategy to combat GBM. To further facilitate the use of p53 mutant GBM cell lines, we created a library of GBM cell lines that stably express clinically important p53 mutations using p53-null LN308 cells to achieve clonal populations of p53 variant cells in a single genetic background. Chemoresistance in cells with mutant p53 occurs by upregulation of Nrf2 expression in many cancers [28,29,30]. To overcome drug resistance, Nrf2 inhibitors may be used to reduce the activity of Nrf2 while sensitizing the cells to chemotherapies [31,32]. For example, treatment of U251-MG and U87-MG cells with FYT720 as an immunosuppressive drug suppressed Nrf2 levels in these cell lines, and significantly sensitized them to TMZ by abolishing the activation of Nrf2 [33]. Our results support the notion that treatment with CET-CH-6 as an Nrf2 inhibitor promotes cellular apoptosis in GBM cells by 30% more than the baseline levels in LN308, independent of p53 status. However, LN308-p53^wt^ showed higher sensitivity to co-treatment using CET-CH-6 and DOX compared to p53-null LN308 cells. In addition, the mutant cells carrying the p53^175^ variant had the highest inhibition in growth, while p53^220^ was the most resistant cell line when co-treated with DOX and CET-CH-6.

Keap1-dependent regulation of Nrf2 in the cytoplasm has been reported as one of the main regulatory mechanisms of Nrf2 [34], although this is not the only possible explanation for Nrf2 activation [35]. Keap1 is expressed in the cytoplasm and mediates ubiquitin-mediated degradation of Nrf2 to tightly control its level in cells. Keap1 is also expressed in the nucleus at a lower level [36]. Upon treatment using CET-CH-6, the expression of Keap1 in the cytoplasm decreased while its expression in the nucleus increased (Figure 3). We also found that, at the molecular level, Nrf2 showed a similar activity pattern to Keap1. Most studies support the translocation of Nrf2 from cytoplasm to nucleus upon redox stress, and the observation of Keap1 following the same path needs further verification. We observed some variations between confocal and western for the nuclear Nrf2 in control cells. For immunostaining, we rapidly fix the cells, and hence there are no transient stresses during the process for the Nrf2 to rapidly move to the nucleus. In contrast, for western, we collect cells after trypsinization. Hence, there are some additional stresses during this process, which causes the translocation of more Nrf2 into the nucleus. Here, we overall compared the nuclear Nrf2 in control cells vs. treated group. In addition, in the confocal images, we are showing a single slice close to the nucleus, but, in western, the result is from the entire nuclear protein. TMZ alone and in combination with CET-CH-6 significantly reduced the expression of Keap1 and Nrf2 in cells in both the cytoplasm and nucleus. This suggests that TMZ causes ubiquitination and degradation of both molecules. 

Cell survival, senescence, and apoptosis signaling regulated by NF-kB, p53, and Nrf2 play important roles in multidrug resistance [37]. Our Western blotting analysis showed that pathways related to p53 such as p21, Puma, BAX, and Bcl2 become highly activated through co-treatment with DOX and CET-CH-6. We observed upregulation of p21 in LN308 with mutant p53 specifically in LN308-p53^175^ and LN308-p53^245^, the two cell lines with very effective treatment results. P21 has been shown to stabilize Nrf2 by binding to Keap1 and preventing its degradation [38]. Puma, which is mainly regulated by p53, showed basal level activation in the LN308 parental cell line, suggesting that there might be pathways involved in the activation of PUMA other than p53-mediated regulation, which also needs further investigation. The level of SOD2 as a downstream target of Nrf2 did not change significantly after treatment with CET-CH-6, except for LN308-p53^wt^, which suggests that the Nrf2 pathway was not completely inhibited in cells carrying p53 mutant variants. Here, we mainly focused on activation of the p53 and Nrf2 pathways. Whether Nrf2 inhibition can enhance activation of the downstream genes of NF-kB needs to be investigated in the future studies. 

Nrf2 has been known to have a dual nature, which raises concerns when considering it as a promising treatment addition [39]. Upon toxicity, a cell will activate Nrf2 to transcribe protective genes and regulate cytoprotective mechanisms. Cells will continuously produce Nrf2 until they reach a threshold. Once the threshold is reached, Nrf2 switches from its cytoprotective functions into apoptotic functions. Our access to a set of small molecules enabled us to pay particular attention to monitoring the switches from protective redox signaling into apoptotic signaling. We expected to see higher apoptotic populations when cells were treated with CET-CH-2 as an Nrf2 activator. However, when we treated U87-MG cells with CET-CH-2, we did not observe any inhibition in cell growth (Figure 5A). This may be because the switching threshold in cells was not achieved by this compound. Unlike U87-MG cells, LN308 cells with mutant p53 showed some response to CET-CH-2 treatment. Co-treatment with CET-CH-2 and DOX showed that CET-CH-2 has some synergistic effects in the inhibition of cell growth, especially on LN308-p53^245^. Interestingly, U87-MG-p53^wt^ and LN308-p53^wt^, even though both had wild-type p53, showed differential responses to treatments with CET-CH-2 or CET-CH-6. In addition to genomic differences between these cell lines, differential responses might be owing to the genomic location of p53. Whereas U87-MG cells possess native endogenous p53, LN308-p53^wt^ possess the same in different genomic locations. 

Currently, TMZ is the first line chemotherapy treatment for GBM. It works by alkylation of cellular DNA leading to DNA cross links and cell death. Nrf2 is understood to be one of the key resistance mediators in GBM and melanoma. Indeed, silencing of the Nrf2 pathway greatly enhances cell death in vitro and in vivo via different pathways [40,41,42,43]. To study the role of our Nrf2 inhibitor on mutant p53 in vivo, we co-treated mice bearing different subcutaneous GBM cells with CET-CH-6 and TMZ. Those findings supported our in vitro results by observing and higher apoptosis in the co-treated mice compared to TMZ alone mice after ten doses of drugs (Figure 8). We also determined that the CET-CH-6 compound is particularly effective on the p53 mutation at amino acid residue 175 which is a clinically common hotspot mutation in GBM, and the most frequently mutated site at the DNA binding region [14]. The p53^R175H^ is a conformational mutant that affects the positioning of L2 and L3, the two amino acid loops that interact with the minor groove of the DNA molecule. p53 in general suppresses the transcriptional activity of Nrf2 and negatively regulates the antioxidant activity to enhance cancer chemotherapy [44]. In contrast, mutant p53, including R175H, with their gain of functions, results in interacting with the Nrf2 to facilitate its nuclear translocation and activation of antioxidant signaling proteins in cells [45]. This process negatively impacts the therapeutic outcome of chemotherapy, with the upregulation of antioxidant signaling. TMZ is particularly effective in patients with MGMT promoter methylation; however, more than 50% of GBM patients retain MGMT expression and demonstrate less sensitivity to TMZ [46]. Further investigations are needed to understand the role of Nrf2 inhibitors in MGMT methylated versus unmethylated tumors. 

## 5. Conclusions

In summary, we introduce a new Nrf2 inhibitor compound that enhances the chemotherapeutic effects on the GBM cell with specific p53 mutations, including the R175H missense mutation. While CET-CH-6 could not completely block the Nrf2 pathway in mutated p53 cells, it activated the downstream proteins of p53 pathways. In the future, this compound can be modified using SAR analysis to synthesize compounds that have improved effectiveness in inhibiting Nrf2 pathways in cancer cells with other p53 mutations. These Nrf2 inhibitors show significant promise for treatment of cancers that carry p53 mutations, when combined with standard chemotherapies.

## Figures and Tables

**Figure 1 cancers-14-06120-f001:**
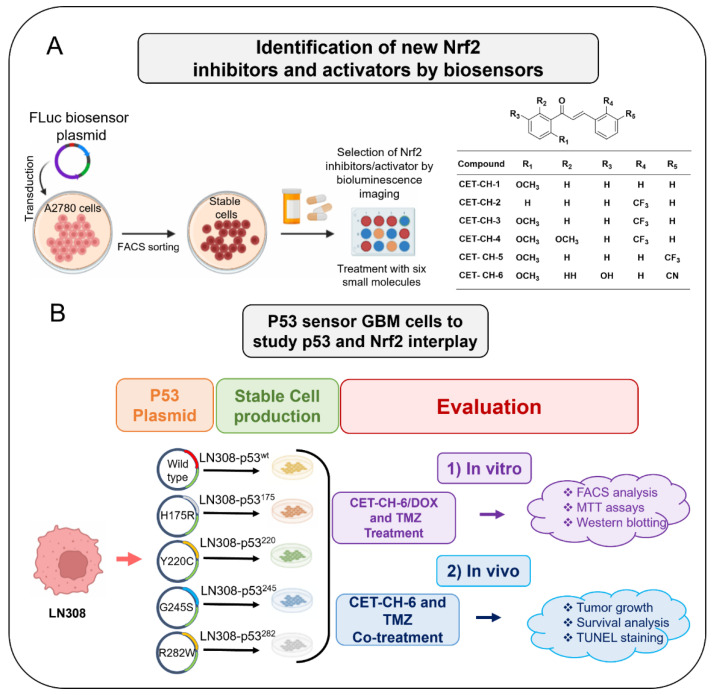
Schematic of study workflow. (**A**) A molecular imaging biosensor identifies Nrf2 inhibitors and activators from a set of small molecule compounds. A2780 cells were transduced by a construct that carries a Firefly luciferase reporter gene under an Nrf2 responsive target gene promoter, the NQO1. Upon activation of Nrf2, the FLuc signal is enhanced, while inhibition reduces the FLuc signal intensity; (**B**) evaluation of Nrf2-mediated mechanism in p53 wild-type and mutant cell lines in response to Nrf2 modulators. LN308 cell line with null-p53 were transduced with five different constructs containing mutant p53 and GFP. In each construct, the mutation was placed in the hotspot structural position in GBM. These cell lines as well as U87-MG with wild-type p53 were then evaluated for apoptotic populations and growth inhibition when co-treated with Nrf2 inhibitor compounds and chemotherapeutic drugs.

**Figure 2 cancers-14-06120-f002:**
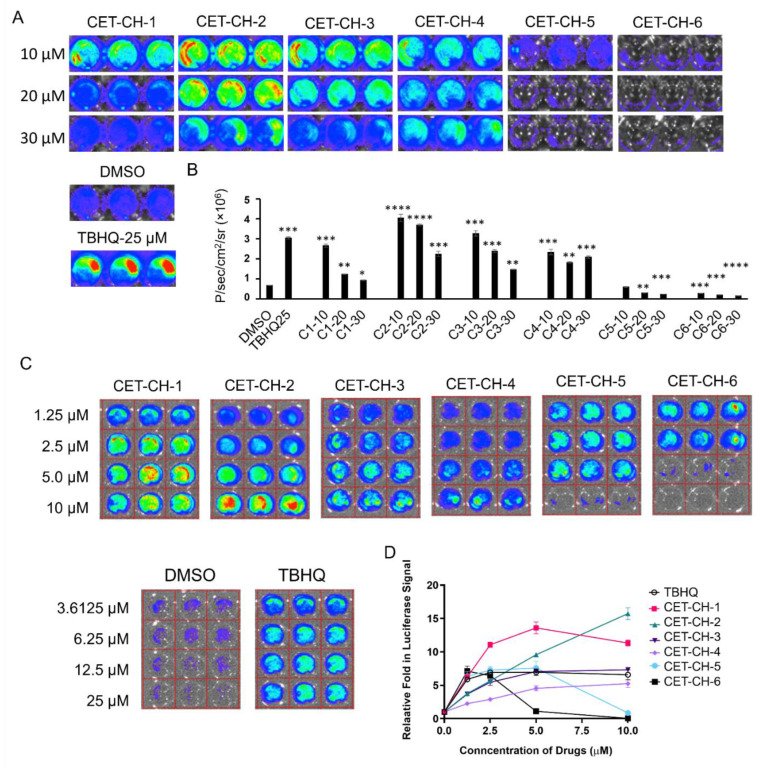
Evaluation of Nrf2 pathway after treatment with six different compounds (CET-CH-1 to CET-CH-6) tested on stably biosensor A2780 ovarian cancer cells. (**A**) Bioluminescence imaging of Firefly Luciferase (FLuc) showing the alteration in Nrf2 pathway in A2780 after treatment with compounds CET-CH-1 to CET-CH-6 at concentrations of 10, 20 and 30 μM, with (**B**) a quantitative plot in which samples are tested for differential expression of FLuc signal in comparison to DMSO as a solvent control sample used for the study. *, **, ***, **** represent *p*-value < 0.05, <0.01, <0.001, <0.0001, respectively. (**C**) FLuc imaging of A2780 cells after treatment with CET-CH-1 to CET-CH-6 at concentrations of 1.25, 2.5, 5, and 10 μM, with (**D**) quantitative plot. In each respective experiment, DMSO and TBHQ images show the minimum (background) and maximum (ROS) bioluminescence signal, respectively. Experiments were performed in triplicate wells. Error bars represent mean ± SD.

**Figure 3 cancers-14-06120-f003:**
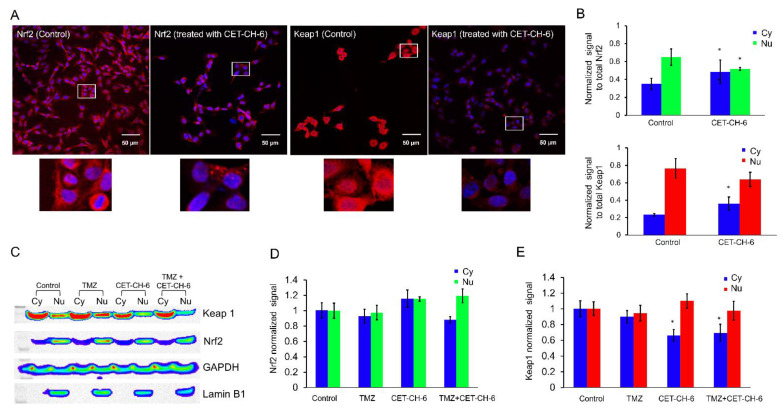
Mechanism of inhibition of Nrf2 in response to treatment with CET-CH-6 in combination with TMZ. (**A**) Confocal microscopy of MDA-MB-231 cells treated with CET-CH-6 and stained with Nrf2 and Keap1 (zoom = 10×). Zoomed in images of the white boxes are shown below each image; (**B**) quantitative analyses of immunofluorescence images shown in (**A**), signals are normalized to the total amount of proteins expressed; (**C**) Western blotting images of MDA-MB-231 cells treated with CET-CH-6, TMZ or their combination for 24 h with quantitative plot of (**D**) Keap1 and (**E**) Nrf2. Protein expressions measured in cytoplasm (Cy) and nucleus (Nu) are normalized to GAPDH and Lamin B1, respectively. All the Western blot images are shown in Appendix A. In (**D**,**E**) the plots are normalized to control Cy and control Nu, respectively. Error bars represent mean ± SD. * represent *p*-value < 0.05.

**Figure 4 cancers-14-06120-f004:**
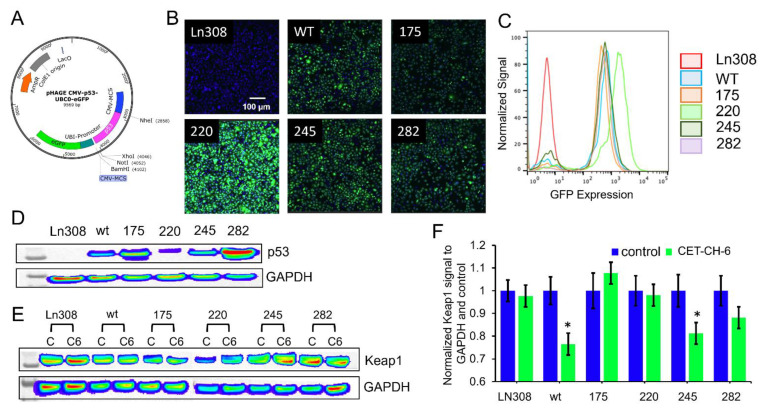
LN308 cell line with no p53 expression engineered to stably express structural p53 variants at different locations. (**A**) the vector designed for the transduction of LN308 cell line to stably express p53 wild-type as well as five different structural mutations in the frequently mutated sites at the DNA binding regions at amino acid residue 175, 220, 245, and 282. These six cell lines were used throughout the study to test the Nrf2 inhibitors and activators; (**B**) confocal microscopy of different clones of LN308 cell lines that stably express green fluorescence protein (GFP) after being transduced with a lentivirus vector constructed for expression of p53 and GFP; (**C**) flow cytometry analysis of different LN308 cell lines stably expressing p53 mutations and GFP; (**D**) Western blotting of LN308 cell lines stably expressing different p53 variants; (**E**) changes in Keap1 expression after treatment of different LN308 cell lines with CET-CH-6 (2.5 μM), with (**F**) quantitative plot normalized to GAPDH and control of each clone. Error bars represent mean ± SD. C: control and C6: CET-CH-6. For whole Western blot membranes, please refer to Appendix A. Keap1 in each cell line is compared to the control signal, and * represents *p*-value < 0.05.

**Figure 5 cancers-14-06120-f005:**
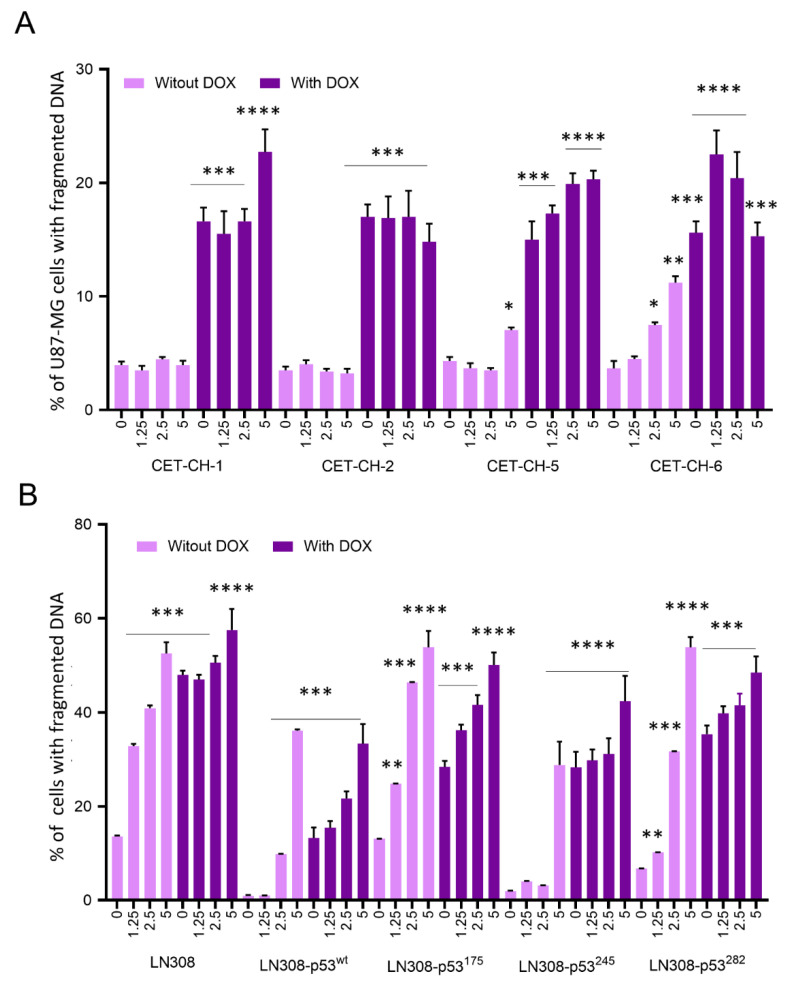
U87-MG cells with wt-p53 expression tested for therapeutic outcome in response to the treatment of Nrf2 activators and inhibitors in the presence and absence of DOX. (**A**) fixed cell population assessed by PI staining based FACS after treatment of U87-MG-p53^wt^ by DOX (0.05 μM) in the presence of two Nrf2 activators (CET-CH-1 and CET-CH-2) and two Nrf2 inhibitors (CET-CH-5 and CET-CH-6), the gating results are shown in Appendix A; (**B**) fixed cell population assessed by PI staining based FACS after treatment of LN308 cells stably expressing different p53 mutations with DOX (0.05 μM) in the presence of CET-CH-6 (concentrations in μM), the gating results are shown in Appendix A. Experiments were performed three times. Cells with fragmented DNA, late apoptotic population, show diffused PI staining compared to non-apoptotic cells. Error bars represent mean ± SD. *, **, ***, **** markers above the columns represent *p*-value < 0.05, <0.01, <0.001, <0.0001, respectively. Each sample is compared to non-treated control sample of the same cell line.

**Figure 6 cancers-14-06120-f006:**
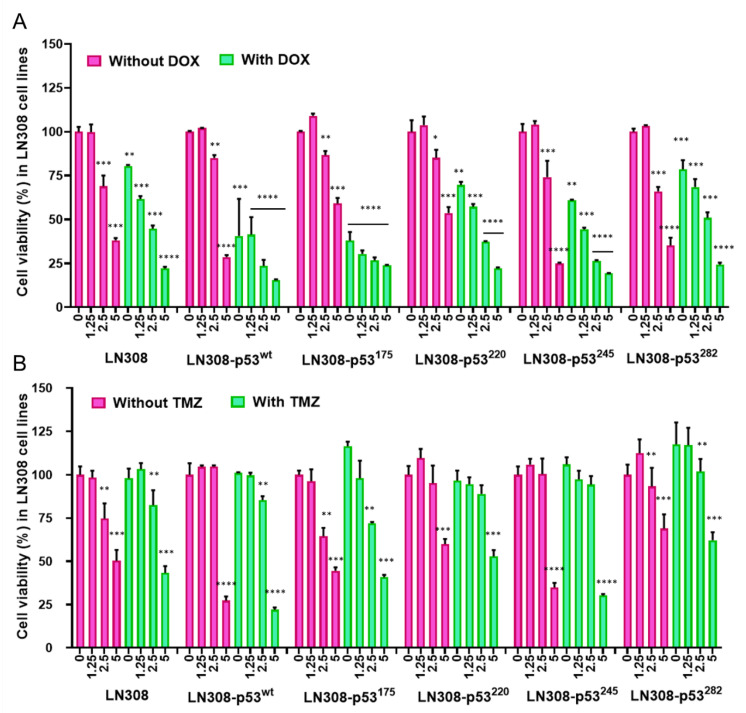
MTT assay result shows that co-treatment of CET-CH-6 with DOX and TMZ inhibits cell growth in LN308 cell lines with different p53 variants. Differential therapeutic responses of LN308 cells stably expressing different p53 variants as assessed using MTT in response to co-treatment with (**A**) CET-CH-6 and DOX, and (**B**) CET-CH-6 and TMZ. Each sample was normalized to the untreated control. Experiments were performed three times, and error bars represent mean ± SD. *, **, ***, **** represent *p*-value < 0.05, <0.01, <0.001, <0.0001, respectively. Samples were assayed 72 h post treatment. Each sample was tested against the untreated control.

**Figure 7 cancers-14-06120-f007:**
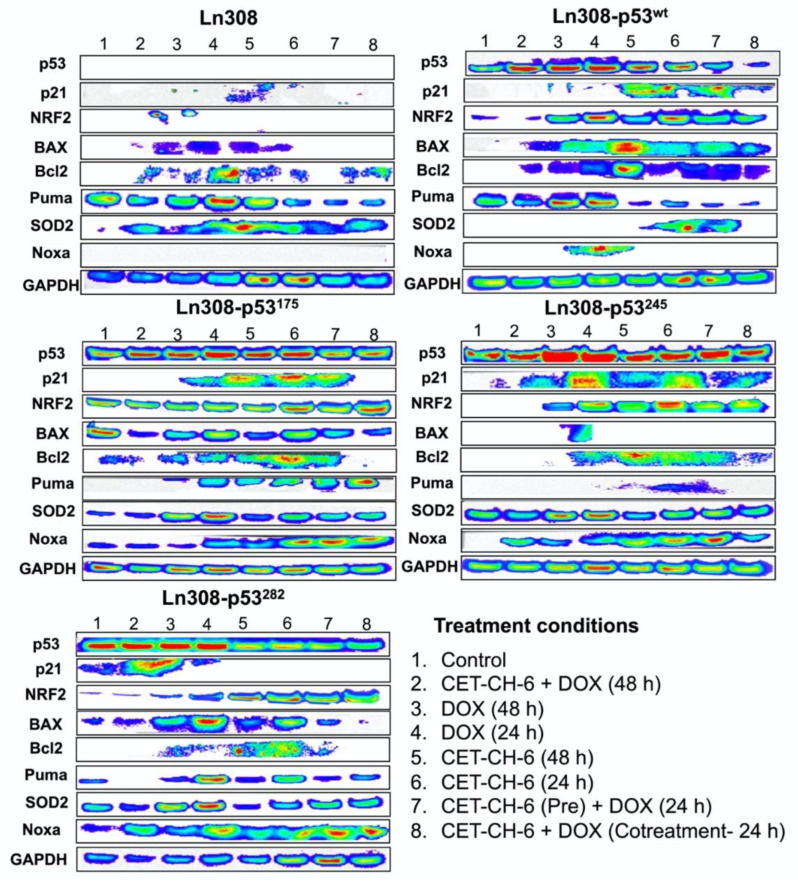
Genes in downstream pathway of p53 and Nrf2 are altered in response to treatment with CET-CH-6 and DOX. Treatment of LN308 cells stably expressing p53 variants (p53^wt^, p53^175^, p53^245^, p53^282^) activated expression of p53 responsive pathway proteins and the Nrf2 pathway. For representative whole Western blot membranes, please refer to Appendix A.

**Figure 8 cancers-14-06120-f008:**
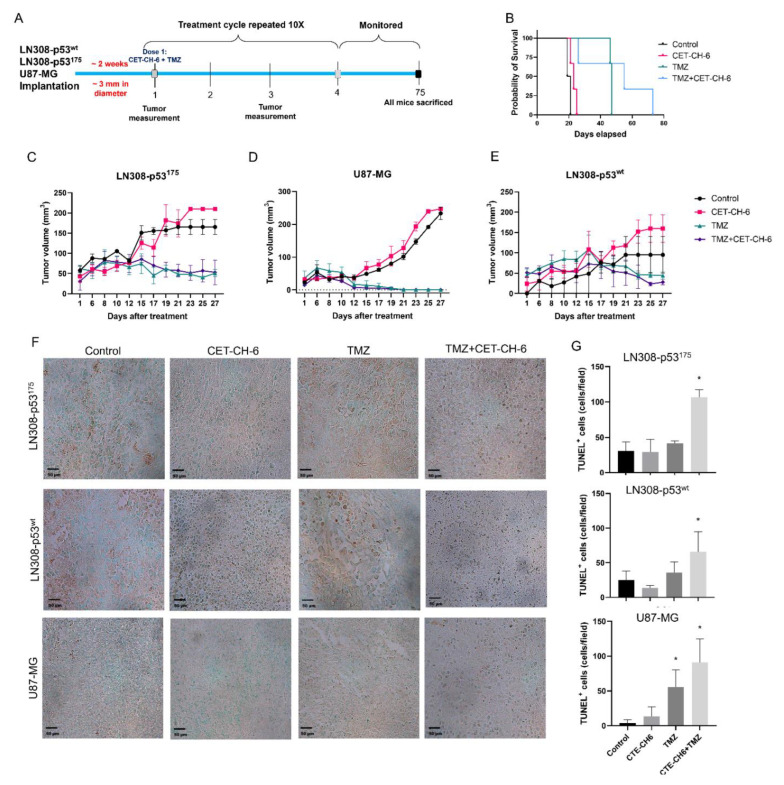
In vivo evaluation of CET-CH-6 in combination with TMZ on NSG mice bearing LN308-p53^175^, LN308-p53^wt^, and U87-MG cells. (**A**) Schematic workflow of in vivo treatment. Mice (N = 4–6/group) were implanted subcutaneously into the left and right flank, respectively. When the tumors reached 3 mm/diameter, the mice were randomized into four groups: control, TMZ (12.5 mg/kg body weight), CET-CH-6 (2.5 mg/kg body weight), CET-CH-6 + TMZ (co-treated with 2.5 and 12.5 mg/kg body weight of CET-CH-6 and TMZ, respectively). The mice received i.p. doses of drugs every four days for ten cycles; (**B**) survival curve of mice with LN308-p53^175^ tumors. Tumor volume measurements for each cell lines over time: (**C**) LN308-p53^175^; (**D**) U87-MG; (**E**) LN308-p53^wt^; (**F**) TUNEL assay to show the apoptotic population in tumor tissue; and (**G**) quantification of tunnel assay, each row shows one cell line. Error bars represent mean ± SD. * markers above the columns represent *p*-value < 0.05.

## Data Availability

The data presented in this study are available in this article (and Appendix A).

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
