# Peer review of "A New Nrf2 Inhibitor Enhances Chemotherapeutic Effects in Glioblastoma Cells Carrying p53 Mutations"

_cancers, 2022, doi:10.3390/cancers14246120_

Round 1
Reviewer 1 Report
The reviewed manuscript describe the development of novel Nrf2 inhibitor, CET-CH-C6, and its effect on glioblastoma cell viability. The investigated compound is supposed to enhance temozolomide and doxorubicin toxicity, in p53 status dependent manner, which in turn affects Nrf2 expression/localization/activity. Therefore, Authors hypothesize that when some mutations occur in TP53 sequence, CET-CH-C6 may be more effective in some cells, but not in the others. Unfortunately, despite multiple constructs and multiple assays, the data are not convincing, and the way of presentation is at least confusing. Paper is too long and lines of evidence provided are more than abstruse. Too much space (and reader’s time) is devoted to the experiments with doxorubicin, and there are attempts to sort out p53-Nrf2 involvement in doxorubicin treated cells. Authors switches from DOX to TMZ, and back, and the reason is not clear. Some attempts to explain the mechanism of drug action, p53 mutants susceptibility, and so on are made with use of doxorubicin. On the other hand, the additive in vitro effect of TMZ and CET6 is lacking in in vitro experiments with TMZ. Then, TMZ appears in in vivo experiments. The opposite is observed for CET6 – it kills cells in vitro and has no effect in vivo.
The experimental evidence quality is not good. Animals number is very low. It is not even known whether the differences in tumor size/survival time between experimental groups reach the statistical significance. The blots quality ranges from moderate to very bad (Figure 6!). It is cumbersome to draw any conclusion based on these data. The sufficient number of repeats and signal intensity measurement is missing from most WB experiments, as well as statistical analysis. There’s no measurement of fluorescence intensity in IF experiments shown in Figure 3A. It is not clear why MD-MBA cells were used here. The effects of CET treatment shown in IF images is different than that shown in WB images. Figure 5 is so overloaded with complex graphs, that it is totally impossible to get the straight message. What is the conclusion we should draw when looking at Figure 5 in regard to TMZ and CET6 synergistic action in LN308 cells?
The link between Nrf2 and p53 R175H is not documented satisfactorily.
It's unclear, why the phrase “p53 hotspot mutation” is used in the manuscript title. Which mutation? Like despite all labor and reasoning one particular mutation cannot be finally specified.
The description of cloning procedure and final constructs carrying p53 mutants is very unclear. Does the final construct contain dTomato? It’s supposed to contain GFP. Were the cells transduced with 2 viruses containing 1. GFP under the control of ubiquitin promoter 2. P53 forms under the control of ubiquitin promoter? At some points transient transfection with pcdna3.1 based constructs is mentioned. Is it not mentioned in the text when transient expression was employed and when lentiviral constructs were used. It’s not clear, why sometimes U87-MG cells are used, and in the other experiments LN308. The different outcomes in in vivo experiments, when different lines are used are not explained.
Moreover, if Authors want to use CET-CH-6 for glioblastoma treatment, showing its ability to cross blood-brain barrier is crucial.
There are also some smaller issues, as follows: 1. The paper on TP53 mutation frequency in GBM is improperly cited. 2. What does it actually mean that “TMZ is a prodrug”? Activated in animal and inactive in cell culture medium?? 3. Does TMZ really “shows effects via epigenetic methylation of DNA and protein in GBM”?
Reviewer 3 Report
To the authors
You present a manuscript that includes a lot of data from many extensive experiments. However, the project described here suffers several shortcomings:
1. inconsistent presentation of results (error bars/ no error bars, lack of statistical analysis for results described as significant)
2. incomplete information regarding the materials and methods used
3. inadequate experimental design (flow cytometry with PI on fixed cells to detect apoptotic population)
Please find my detailed comments below:
please use traditional black/white presentation for western blots.
The manuscript would benefit from a focus on the most important components. For example, after Figure 2, only CH-6 should be focused on.
· Define EB6 abbreviation in Figure 1B. I don’t find a reference to it anywhere else in the manuscript.
· Explain choice of A2780 cells
· Why is it necessary to use MDA-MB-231 cells that overexpress Nrf2 for the experiments shown in figure 3? Why not use more relevant LN308?
· “We also observed higher levels of Nrf2 and Keap1 in the nucleus than in cytoplasm after treatment with CET-CH-6 (Fig. 3A)”. Your representative images do not show this. Please quantify average nuclear and cytoplasmic Nrf2 and Keap1 intensities via imageJ to demonstrate this. The IF images also need a scale bar.
· Please include catalog numbers and manufacturer information for all antibodies. This is currently inconsistently done. Santa Cruz for example makes 3 different Keap1 antibodies.
o “…and SOD2 primary antibody from Santa Cruz Biotechnology (SC 137254)”
o “We stained the cells using Nrf2 and Keap1 antibodies (1:200) diluted in PBST with 1% bovine serum albumin and incubated them overnight in a humidified chamber.”
· In the intro you describe Nrf2 as ‘maintianed in the cytoplasm’ and translocated to the nucleus after stress. Your IF control images show mostly cytoplasmic Nrf2 (3A) but your western blot control shows mostly nuclear Nrf2. Please explain. I recommend the inclusion of a nuclear fraction marker protein (Lamin B1, H3, Nucleolin etc.)
· How many independent repeats were done for the experiments in figure 3. Please show statistical analysis of densitometry (Error bars and potential significant differences in figures 3C and D). You do show error bars in the next figure
· Figure S2 is first mentioned towards the end of the manuscript. Please Re-order supplemental figures in order of mention.
· Figure S4: this experiment is not suitable to show apoptotic population. First, the sample preparation is inadequately described in the methods (I assume the cells were fixed/permeabilized in some manner). PI only enters and stains dead cells. To show apoptosis, an Annexin5 / PI staining experiment of unfixed, live cells is appropriate. Second, your PI staining presumably gives you the classic G1-peak, S-phase bridge, G2/M peak cell cycle profile. What’s the explanation for the peaks or shoulders after the G2/M peak? Please show your gating of your flow reads in the supplements and re-do these experiments using an Ax5/PI apoptosis assay. Same goes for following similar flow experiments.
· Also, why use U87-MG GBM but not the LN308 cell lines and DOX, but not TMZ for experiments in Figures S4, S5?
· Figure S6 is an inappropriate experimental design, similar to Figure S4. Use Ax5/PI
· Results of figures S4, 5, 6 need to be summarized (graphs), statistically analyzed and added as an actual figure into the paper.
· All MTT results: Triplicate wells are technical replicates and not independent repeats of experiments. Independent experiments are done on separate plates and at separate times. And p-values between technical replicates are not valid.
· In addition to your bar graphs of Figure S5 and Figure 5, Figure S7, Show results of all MTT assay experiments in the form of annotated pictures of the 96-well plates after formazan solubilization. Also, in my opinion, MTT assay result graphs should always come in the form of dose-response curves with calculated IC50. The main figure only needs the 72h results, the rest can go into supplements.
· Results of figure S8 are irrelevant. Delete.
· Dose response curve experiment in Figure S9 is inadequate. Only 1 datapoint is within the dynamic range. Readjust doses and repeat experiments with more datapoints within dynamic range.
· Drug synergy is discussed but not calculated. Please show synergy by calculating it (Excess over Bliss, CalcuSyn, COMBENEFIT, etc.). Additivity can not be excluded from the data that is presented.
· Figure 6: results should be summarized and statistically analyzed using densitometry. Why is Keap1 not included here? What was your hypothesis for this experiment? Which results did you expect and which results were unexpected?
· What are the dotted lines in figure 7C? Is survival meaningful, with 2 different tumor types implanted in each mouse? Where is the statistical analysis?
· Figure 7D, please quantify and indicate apoptotic cells in the representative images
· “In combination with TMZ, CET-CH-6 significantly delays tumor growth”. Please show the data for this. There is no direct comparison with the stat. sig. calculated.
· Figure S11B: please show absolute tumor volumes, and C: what are the dashed lines. Again, you say the combination treatment was significantly more effective. Then please show the direct comparison of tumor volume + corresponding statistical analysis (for tumor volume and survival)
All the best!
Reviewer 4 Report
It is noted in the manuscript that the authors have done extensive work and their results are interesting. However, I believe that major revisions must be made to be published.:
In order to affirm that the process of regulated cell death is apoptosis, it is not enough with a single technique, such as TUNEL, they should perform at least two more, one of them involving a molecular target of said mechanism (i.e. activation of caspases or an inhibitor of apoptosis).
Statistical information is missing in the description of the results and in most figure captions. P-values ​​should be indicated in the text each time a statistical comparison is referenced. These values ​​should also be indicated in the figures.
In M&M section, it should be included how the nuclear and cytoplasmic extracts were obtained, because what I read in the protocol that refers to the WB, only the obtaining of total extracts is mentioned.
In the M&M section: "2.11. FACS analysis of PI staining for measurement of apoptosis", the authors describe a commonly used PI staining to assess cell cycle. However, they argue that they use the technique to discriminate living cells from dead cells, but they never mention that they evaluate the sub-G1 peak, which, if not present, does not exclude the induction of apoptosis. Could the authors justify how they proceeded? Also, because the cells are fixed and labeled with PI, they are no longer alive.
In Figure 5, the authors talk about apoptosis and plot viability. What technique are they using: MTT or PI staining?
Round 2
Reviewer 3 Report
I thank the authors for addressing many of my comments. Unfortunately a major issue remains: The authors do not sufficiently demonstrate the induction of apoptosis. The flow experiments are simply insufficient. I would have expected that the authors take this particular criticism more seriously, as it was raised by at least 3 of the 4 reviewers. Propidium Iodine stains dead cells, and a weakened PI signal shows degradation of DNA. This may stem from apoptosis, but is by no means guaranteed. Use AnnexinV or any other widely accepted apoptosis assay (cleaved caspases, etc.). I will request a major revision of this issue and expect it to be addressed in full.
Round 3
Reviewer 3 Report
I thank the authors for taking many of the reviewers suggestions seriously.
Reviewer 4 Report
I attach my final answer.Best regards.
